# Transfer-Learning-Based Temperature Uncertainty Reduction Algorithm for Large Scale Oil Tank Ground Settlement Monitoring

**DOI:** 10.3390/s24010215

**Published:** 2023-12-29

**Authors:** Tao Liu, Tao Jiang, Gang Liu, Changsen Sun

**Affiliations:** College of Optoelectronic Engineering and Instrumentation Science, Dalian University of Technology, Dalian 116024, China; liutao2020@mail.dlut.edu.cn (T.L.); jiangtao19980125@mail.dlut.edu.cn (T.J.); newgun@mail.dlut.edu.cn (G.L.)

**Keywords:** optical fiber sensor, ground settlement monitoring, temperature uncertainty, artificial neural network, transfer learning method

## Abstract

Sensors operating in open-air environments can be affected by various environmental factors. Specifically, ground settlement (GS) monitoring sensors installed in oil tanks are susceptible to non-uniform temperature fields caused by uneven sunshine exposure. This disparity in environmental conditions can lead to errors in sensor readings. To address this issue, this study aimed to analyze the impact of temperature on GS monitoring sensors and establish a mapping relationship between temperature uncertainty (fluctuations of measurement caused by temperature variation) and temperature variation. By collecting the temperature information and inferring the temperature uncertainty being introduced, this interference can be removed. However, it is crucial to note that in real-world complex scenarios, the relationship between temperature uncertainty and temperature variation is not always a constant positive correlation, which limits the data available for certain periods. Moreover, the limited availability of data presents a challenge when analyzing the complex mapping relationship. To overcome these challenges, a transfer-learning-based algorithm was introduced to develop a more accurate model for predicting temperature uncertainty based on temperature variation, even with limited data. Subsequently, a practical test was conducted to validate the proposed algorithm’s performance. The results demonstrated that the algorithm outperformed a simple linear fitting model using the least squares method (LSM), achieving an improvement of up to 21.9%. This outcome highlights the algorithm’s potential for enhancing the performance of GS sensors in daytime monitoring and contributing to the safe operation of oil tank facilities and infrastructure health monitoring.

## 1. Introduction

Ground settlement in oil tanks is a critical factor that can compromise their structural integrity and service life [1,2,3,4], as depicted in Figure 1a. This emphasizes the importance of prioritizing GS monitoring for large scale infrastructures. Currently, remote sensing methods, such as the use of Total Station, are commonly employed for GS monitoring, but they are susceptible to environmental interference and are labor-intensive [5]. Until now, there has been a lack of practical online GS monitoring techniques. Our research group initially proposed an online GS monitoring method that combines optical low-coherence interferometry with a hydrostatic leveling system (HLS), and this has made significant progress. However, one major challenge being faced is the interference caused by non-uniform temperature fields [6]. During the daytime, solar irradiation can be impeded by the large size of the oil tank, resulting in significant temperature differences of up to ${30\;}^{o}C$ between the sunlit and shaded areas in the summer [3,6]. This temperature difference can impact the HLS directly and lead to considerable temperature uncertainty (large fluctuation of GS measurement). As depicted in Figure 1b, the practical continuous GS monitoring results over a week revealed significant fluctuations (reaching centimeter scales!!!) in the readout of the optical GS sensor. Notably, during daytime, the sensor readout exhibited a drop, while it returned to its original levels during nighttime, implying the effectiveness of the sensor’s performance during nighttime operations. Consequently, it is imperative to delve deeper into the underlying cause of this phenomenon and thoroughly investigate its mechanisms.

The GS sensor measures the distance between the top of the sensor container and the liquid level, to obtain GS information based on HLS, as has been illustrated in detail in Ref. [7]. When the temperature increases, thermal expansion takes place, causing a decrease in the measured distance. This is the reason behind the observed pattern. Given the distribution of sensors around a large oil tank (particularly one of a $10,000\;m^{3}$ scale with dimensions of 21.8 m in height and 80 m in diameter), it would not be economically feasible to construct a large shelter or implement other hardware-based improvements to achieve a uniform temperature field for sensors by building an air-conditioned environment [8] or utilizing thermal insulation materials [9].

Therefore, in order to address this issue, we have made the decision to tackle it through the implementation of software algorithms. The problem thus becomes one of analyzing and understanding the mapping relationship between the temperature and temperature uncertainty. As illustrated in our previous publication Ref. [6], the temperature can directly influence the GS measurement result in daytime. Furthermore, the GS measurement results at night can be considered as the ground truth results, because the temperature environment is uniform without sunshine. The difference between GS measurements in daytime and nighttime can be considered as the temperature uncertainty induced by temperature variation. If we can build a mapping relationship between the temperature uncertainty and temperature difference, then when the temperature information is collected, the temperature uncertainty can be calculated and removed. This means that the GS measurement can be conducted in daytime.

Theoretically, as the temperature increases, the liquid expands. This expansion causes the liquid level to rise and leads to uncertainties in GS measurements. That means that there exists a positive correlation between the temperature uncertainty and the temperature variation in each sensor. However, several factors—such as the large scale of the HLS system, installation errors, uncontrolled liquid convection, and heat transfer along the connecting tube—can make the relationship between the temperature uncertainty and temperature variation more complex [10,11]. Furthermore, the thermal expansion in the HLS system can vary over time, leading to a complex and time-dependent relationship between temperature uncertainty and temperature variation. This indicates the relationship can be stable at certain times but may change over time, causing limited data availability. On the other hand, GS is a slow-varying process, and we can set the sampling frequency to once every hour or once every two hours to record the GS variation. A higher sampling frequency is not necessary or required in this context. These two factors mean that only few data are available in a certain time period; these are not enough to train a stable convergent artificial neural network (ANN).

To overcome the challenges posed by limited data and the complex mapping relationship between temperature uncertainty and temperature variation, the transfer learning method combined with ANNs is introduced. In recent years, ANNs, a type of machine learning technique, have shown great promise in dealing with complex mapping relationships. In this research, a type of ANN called a back propagation (BP) neural network is employed to infer the temperature uncertainty and exclude it.

However, the shortage of data can present another obstacle in achieving accurate predictions. Thus, the transfer learning method is applied, which involves a pre-training process with similar (but not identical) data which are specifically created for the task. In the pre-training stage, the roughly positive correlation relationship mentioned in Ref. [6] is used; then, the practical data are incorporated into the training process. This approach allows us to effectively address the complexity of the relationship between temperature uncertainty and temperature variation while mitigating the data shortage. To validate the effectiveness of our algorithm, a practical test was conducted, comparing its performance with an LSM. The results showed an improvement of approximately 21.9% with our proposed transfer learning-based algorithm, indicating the good generalization ability of the neural network.

In summary, our research focuses on addressing the temperature uncertainty problem in the optical GS sensors used for oil tank monitoring. By utilizing transfer learning and incorporating practical temperature information, the data requirements for training the ANN are reduced and the trained neural network can capture the complex relationship between temperature uncertainty and temperature variation effectively via practical validation.

## 2. Theoretical Background 

### 2.1. GS Monitoring Sensor Based on Low-Coherence Interferometry and HLS

In oil tank safety monitoring, GS monitoring is of vital importance for the thin shell structure of oil tanks, to prevent severe damage such as distortion or cracking of the tank walls [4,12,13]. The prerequisite requirement of GS monitoring is to establish a stable reference point for comparison with the measurements obtained at various test points.

In response to the requirement, researchers have proposed solutions based on the HLS to effectively conduct GS monitoring. The HLS rule comes from the Bernoulli’s law, which is presented in the following form:(1)$$\frac{1}{2}\rho v^{2}+P+\rho gh_{i}=const$$
where v is the liquid flow speed, P is the hydraulic pressure, ρ is the liquid density, g is the acceleration of gravity, $h_{i}\;is\;the$ height of the liquid column of container i, and i is the index of the container. Considering the working condition of liquid—which is always in resting state (v=0)—and sensors that work in one common workplace where P is constant and equal everywhere, Equation (1) can be simplified as
(2)$$\rho gh_{i}=const$$

The HLS holds significance as it enables the determination of vertical displacements with a high accuracy of up to 0.01 mm [14]. Consequently, HLS is primarily employed for highly precise measurements in engineering object monitoring with stable environment, such as underground tunnels [15], dams [16], etc. In order to get highly precise and reliable results, the corrections that cause the change of height of liquid column $h_{i}$ in HLS should be considered when interpreting the observational outcomes. 

As shown, the application of HLS allows for the establishment of an equivalent reference point, akin to the liquid surface. The HLS was established as shown in Figure 2a, by converting the GS of each sensor into the distance from the top of sensor container to the liquid level for calculation. 

The relative settlement can be calculated as below, which is elaborated in more detail in Ref. [7]: (3)$$H_{i}\;–\;H_{ref}=\left(K_{i}^{\prime }-K_{i}\right)-(K_{ref}^{\prime }-K_{ref})$$
where $H_{i}$ is the GS of the oil tank, as measured by the installed sensor around the oil tank evenly and indexed by i (corresponding to the sensor locations); $H_{ref}$ is the GS of the reference sensor; $K_{i}$ is the distance from the top of the sensor probe to the initial liquid level inside sensor i; and $K_{i}^{\prime }$ represents the distance from the top of the sensor probe to the final liquid level inside sensor i after the GS occurred. $K_{ref}$ and $K_{ref}^{\prime }$ are counterparts of the reference sensor. The reference sensor is fixed on a stable benchmark and, in practice, $H_{ref}$ is conveniently ascribed to zero by subtracting its initial value. By collecting the measured data from all sensors, the overall GS of the oil tank can be evaluated. The pivotal aspect of conducting GS monitoring lies in determining the changes in the liquid level from the top of the container. To realize high-accuracy measurements, low-coherence interferometry is utilized, which offers extremely high optical precision, to facilitate the acquisition process.

The utilization of low-coherence interferometry offers an extremely high optical precision of up to 50 um, to facilitate the acquisition process. In Figure 2b, the sensor probes are connected to the optical scheme. The main derivation for low-coherence interferometry was copied from Ref. [2], as
(4)$$I\left(Q\right)=I^{\left(1\right)}\left(Q\right)+I^{\left(2\right)}\left(Q\right)+2\sqrt{I^{\left(1\right)}\left(Q\right)}\sqrt{I^{\left(2\right)}\left(Q\right)}\gamma_{12}^{r}(\frac{L_{AB}-L_{CD}}{c})$$
where $\gamma_{12}^{r}$ is the real part of the complex degree of coherence $\gamma_{12}$; $I^{\left(1\right)}\left(Q\right)$ and $I^{\left(2\right)}\left(Q\right)$ are the optical intensities from the two interference paths, respectively; and $L_{AB}$ and $L_{CD}$ are the optical paths between mirrors A and B and mirrors C and D, which are represented by red arrows in Figure 2b above. The optical principle is shown in Figure 2b. The low-coherence light source has a central wavelength at 1310 nm and a full-width-at-half-maximum (FWHM) of about 45 nm. The light source is connected to port 1 of the first circulator. After leaving port 2, the light is reflected by half-reflection mirror A and total-reflection mirror B. Mirror B is mounted on a stepping motor, and the light is projected through a collimated grin lens. Port 3 of the first circulator is connected to port 1 of the second circulator. The light from port 2 is half reflected by mirror C and half coupled to a collimated grin lens D, by which the light is collimated as a parallel beam and projected to the liquid surface. The 1 × 16 optical switch can change the main optical path to different sensor probes individually; it is controlled by a computer. The light from port 3 is detected by a photodetector (PD) and converted into an analog electrical signal. Then, the data acquisition board (DAQ) can convert this signal into a digital signal. When the electrical motor moves and satisfies the equal optical path difference condition as $L_{AB}=L_{CD}$, a peak signal can be found in the interference pattern. Using this, the liquid level change can be continuously recorded, and the GS can be deduced [3].

Low-coherence interferometry provides a high accuracy of displacement measurement of within 50 um, which can be considered negligible in practical applications. However, it is crucial to recognize that the primary source of errors in the GS monitoring system lies within the HLS itself. In particular, the variations in liquid heights are influenced by thermal expansion caused by temperature fluctuations. This discrepancy in liquid level due to temperature variations is the underlying cause of temperature-related errors in GS monitoring. As illustrated in Figure 2c, the unbalanced sunshine produces a non-uniform temperature field, leading to temperature uncertainty of each sensor. The sensors are strategically installed around the tank according to the illustration provided, and the solar irradiation is as shown. Sensors #1 to #8 are placed around the tank, while sensor #9 is positioned at a distant location away from the tank. During instances of solar irradiation on the oil tank, the sensors located at the back tend to experience lower levels of sunshine, consequently leading to lower temperatures recorded by these sensors. Conversely, the sensors situated at the front of the tank are exposed to more direct sunlight, resulting in higher temperatures being measured by these sensors.

### 2.2. Artifical Neural Network

ANNs, inspired by the structure and functioning of the human brain, are computational models that can learn and make predictions based on input data. They are widely used in various fields, including science and technology applications such as computer vision [17], natural language processing [18], medical care [19], and data mining and predictive analysis [20] for their ability to handle complex patterns and relationships in data processing. 

ANNs consist of interconnected nodes called neurons, organized in layers. The input layer receives data, which is then processed through hidden layers that perform computations. The output layer generates predicted results. The connections between neurons have associated weights that determine the importance of each input. During the training process, the network adjusts these weights using optimization algorithms such as gradient descent to minimize the discrepancy between its predicted output and the actual output.

One of the key advantages of neural networks is their ability to capture complex patterns and non-linear relationships in data handling. This makes them powerful tools for prediction, classification, and regression tasks. By learning from large amounts of labeled data, neural networks can generalize and make accurate predictions on unseen data. 

In this research, ANNs are introduced for prediction, to leverage their capability of extracting meaningful features from data and modeling complex relationships. By training the network on available data, it can learn patterns and make predictions based on new inputs. This approach is beneficial when traditional methods struggle to capture underlying patterns or when the relationships are non-linear in nature. Specifically, the BP neural network is applied for its nonlinear mapping capabilities, self-learning and adaptability, and generalization ability [21]. The basic structure and calculation process of this network are introduced below.

A typical three-layer BP neural network topology is shown in Figure 3. The network consists of an input layer, a hidden layer, and an output layer, with each layer comprising multiple neurons. In the diagram, each node represents a neuron. The number of nodes in the input and output layers is determined by the characteristics of the data, while the number of nodes in the hidden layer can be adjusted as per requirements. The function of the neural network is to predict unknown attributes based on known attributes of the samples.

Figure 3 illustrates an example in which there are three attributes represented by nodes in the input layer and two unknown attributes represented by nodes in the output layer. The connections between nodes represent the weights, and these connections between layers form the weight matrices, denoted as $W^{\left(1\right)}$ and $W^{\left(2\right)}$, which govern the calculation process between layers. Suppose there is one sample containing three elements: $i_{1}$, $i_{2}$, and $i_{3}$. The input layer can be written as a matrix I as follows:(5)$$I=\left[\begin{array}{c} i_{1}\\ i_{2}\\ i_{3} \end{array}\right]$$

The hidden layer matrix H is calculated by
(6)$$H=g(W^{\left(1\right)}\cdot I-A)=g\left(\left[\begin{array}{ccc} w_{1,1}^{\left(1\right)} & w_{1,2}^{\left(1\right)} & w_{1,3}^{\left(1\right)}\\ w_{2,1}^{\left(1\right)} & w_{2,2}^{\left(1\right)} & w_{2,3}^{\left(1\right)} \end{array}\right]\left[\begin{array}{c} i_{1}\\ i_{2}\\ i_{3} \end{array}\right]-\left[\begin{array}{c} a_{1}\\ a_{2} \end{array}\right]\right)=\left[\begin{array}{c} h_{1}\\ h_{2} \end{array}\right]$$
where A is the bias matric, and g is the sigmoid function, defined as
(7)$$g\left(x\right)=\frac{1}{1+e^{-x}}$$

Similarly, the output layer consists of two elements, which can be written as matrix O:(8)$$O=\left[\begin{array}{c} o_{1}\\ o_{2} \end{array}\right]$$

The output layer matrix O is calculated by
(9)$$O=g(W^{\left(2\right)}\cdot H-B)=g\left(\left[\begin{array}{ccc} w_{1,1}^{\left(2\right)} & & w_{1,2}^{\left(2\right)}\\ w_{2,1}^{\left(2\right)} & & w_{2,2}^{\left(2\right)} \end{array}\right]\left[\begin{array}{c} h_{1}\\ h_{2} \end{array}\right]-\left[\begin{array}{c} b_{1}\\ b_{2} \end{array}\right]\right)$$
where B is the bias matrix calculated with nonlinear function g(x).

Then, the weight matrices are modified to ensure that the predicted results are sufficiently close to the results desired by the training process. The total error E in the performance of the network is defined as
(10)$$E=\frac{1}{2}{\left\|O-O_{p}\right\|}^{2}$$
where O denotes the desired output results matrix. To minimize E using gradient descent, it is necessary to compute the partial derivative of E for each weight in the network. Since the structure of the neural network is often complex, directly calculating the partial derivatives can be computationally intensive. To overcome this limitation, the BP neural network employs an error inverse propagation algorithm based on the chain rule [18]. This algorithm utilizes the network structure to calculate the partial derivatives from the output layer to the hidden layer and then to the input layer. The weight matrix $W^{\left(2\right)}$ and bias matric B are modified based on the gradient between the output layer and the hidden layer. Subsequently, the weight matrix $W^{\left(1\right)}$ and bias matric A are modified based on the gradient between the hidden layer and the input layer. This modification process iterates until the desired level of accuracy is achieved. Once trained, the neural network can be utilized to predict the unknown attributes of the samples.

In application, the designed ANN is much larger than the exemplified BP neural network, and it contains one input and one output. The input is the temperature difference and output is the predicted temperature uncertainty. Our task is to process and utilize data to train the ANN.

### 2.3. Transfer Learning Fundamentals

Although ANNs have achieved remarkable success in various fields, a crucial requirement for their effective utilization is the availability of sufficient labeled data that align with the same feature space and distributions for training and testing. However, collecting an ample amount of labeled data can often be prohibitively expensive and, in practical conditions, even unattainable. To overcome this challenge, transfer learning has emerged as a promising solution. Transfer learning can learn the complex implicit relationships between input and output data using only a small amount of experimental data [22,23]. 

In this section, a brief overview of the transfer learning method and the engineering background are presented. In a previous publication (Ref. [6]), there is a roughly correlative relationship between temperature uncertainty and temperature variation for a specific sensor, although this correlation and its corresponding coefficient vary over time. Consequently, in a certain time period, only a limited amount of data may be available to effectively grasp and understand this relationship. For instance, in scenarios where there is a 24 h time period with an hourly sampling rate, the 24 data per day are not enough to train an ANN to converge. In practice, at least hundreds of data are required to produce a convergent ANN, and more are needed to grasp a more complex relationship. To tackle the issue of insufficient labeled data collection in practical experiments, we propose a method based on the concept of transfer learning. This approach aims to leverage prior knowledge from related domains or tasks to enhance the performance of the ANN and overcome the limitations posed by the scarcity of labeled data. In the source domain $D_{s}$, the training data used are simulated correlative data; in the target domain $D_{T}$, we use the practical data, as illustrated separately below.

Figure 4 presents a schematic representation of how transfer learning is employed to reduce temperature uncertainty via coefficient transfer. The main objective of transfer learning is to enhance the performance of prediction models in $D_{T}$ by leveraging knowledge from a different but related Ds.

In the source domain on the left-hand side, a pre-trained model is trained using simulated data. These simulated data are generated based on recent data and simulate a linear relationship between temperature uncertainty and temperature variation. As mentioned above, the temperature uncertainty is roughly correlated with temperature difference, as has been proved in our previous research in Ref. [6]. Based on this, the roughly linear relationship is used as a reference for $D_{s}$. For application, a large dataset was created and simulated with the linear relationship, which can satisfy the demands of large-scale datasets. In theory, it is important to note that the pre-trained dataset is different from the target domain but still has relevance. By applying this large dataset, the pretrained ANN can be trained with the coefficients being calculated.

In the target domain on the right-hand side, the pre-trained model’s weight matrix (Ws) and bias matrix (Bs) are fine-tuned during the later training process using limited practical experimental data. This allows for the updating or fine-tuning of the model’s learned parameters during training. As a result, the model can exhibit better predictive performance in our real measurement environment. To achieve this, we utilized the transfer learning method to pretrain the model on simulated data and then transfer the learned implicit relationship to a small amount of experimental data for fine-tuning the model. This approach reduces the reliance on a large quantity of experimental data for training, while still maintaining accurate predictions of temperature uncertainty with a good generalization ability of the neural network.

## 3. Temperature-Uncertainty-Reduction Algorithm

In this section, we present an algorithm based on transfer learning to mitigate the temperature uncertainty observed in the daytime GS readouts, as depicted in Figure 1b. The introductions provided here offer background information and an overview of the process.

### 3.1. Engineering Background

Under ideal conditions with a consistent temperature field, the sensors only respond to the settlement or uplift of the tank position. However, as mentioned above, the temperature can induce temperature uncertainty in the measured results. Our proposed algorithm aims to reduce this uncertainty in GS measurements. 

The fundamental concept is to measure the temperature and calculate the uncertainty being induced by it. The key aspect is to establish a model that relates the temperature difference to the resulting temperature uncertainty. The temperature information is collected according to the reading from the distributed Raman temperature measurement system. Then, the temperature uncertainty is treated as the GS measurement result in daytime minus the counterpart at night. The temperature variation is calculated similarly to the temperature uncertainty. 

After analysis, it is found that the relationship between the temperature difference and variation can vary differently between different sensor probes. Therefore, an individual model is established for each sensor to relate the temperature uncertainty and temperature difference separately. After establishing the model and temperature information, the uncertainty can be calculated and removed from the raw GS measurement data. 

### 3.2. Algorithm Flow

The basic structure of our model consists of three components: pretrained model, training model, and temperature uncertainty reduction, as illustrated in Figure 5. 

In the first component, the data collected over a continuous period (preferably from the closest neighbors, spanning 72 h) are utilized. These recorded data are then subjected to LSM to establish a linear regression model. To augment the dataset, an additional set of 10,000 data points is generated through extrapolation based on the linear relationship. This new dataset is divided into a training dataset with 7000 data points and a testing dataset with 3000 data points, for a 70:30 training: validation split. The neural network model used in our research consists of 1 input layer, 10 hidden layers, and 1 output layer. The input layer receives the temperature variation as input, and the output layer produces the corresponding temperature uncertainty as output. 

Next, in the subsequent training phase, the neural network undergoes fine-tuning using data from the most recent 24 h period. However, the limited size of the dataset is not sufficient to fine-tune the coefficient of the network; hence, a strategy is employed to augment it by replicating each data point 10 times, resulting in a dataset of 240 data points. All the data points are then used in the training stage. It is important to note that simply replicating the raw data to increase the dataset size is not sufficient for training an ANN. The raw data often contain significant random noise, which can hinder the ANN’s convergence and lead to inaccurate predictions with considerable uncertainty. Therefore, employing transfer learning becomes imperative to enhancing the ANN training process and mitigating these challenges.

In the final part, the temperature uncertainty reduction is conducted. When new data are collected, the temperature variation of each sensor is fed into the network to predict temperature uncertainty. The predicted temperature uncertainty ($H_{pred}$) is then subtracted from the GS readings ($H_{raw}$) to calculate the real GS ($H_{real}$), effectively removing the temperature interference, as
(11)$$H_{real}=H_{raw}-H_{pred}$$

It is worth noting that the transfer learning model is conducted through parameter transfer. The approximately similar data tendencies are utilized for pre-training the network; then, we enhance the generalization ability of the neural networks by incorporating practical data. This process is crucial for capturing complex relationships, resulting in more accurate temperature uncertainty predictions and better temperature uncertainty reduction. 

## 4. Practical Test and Data Comparison

To evaluate the performance of our proposed algorithm, a practical experimental test was carried out. The proposed algorithm was compared with the formal LSM method presented in our previous publication [21]. By conducting this experimental test and data comparison, the reliability and effectiveness of the proposed algorithm for addressing temperature-induced errors in oil tank GS monitoring systems are validated.

### Data Comparison

The temperature information was measured by a distributed Raman temperature sensor (HG-DTS-160, Tianjin Huigan Optoelectronic Technology Co. Ltd., Tianjin, China). Its temperature measurement range was about –40–${100\;}^{o}C$, with an accuracy of $\pm \;{1\;}^{o}C$ and a resolution of ${0.1\;}^{o}C$. The GS sensor had a resolution of 0.1 mm, as introduced in detail in Ref. [6].

Here, the algorithm-processed data and those calculated with LSM are compared. Theoretically, for such a short time period (24 h), the GS of a large oil tank is negligible without loading condition change. From Figure 1b, the readout at midnight shows the change is limited to within 3 mm for 1 week. Besides this, actual tests have demonstrated that in a loading process, the GS between the totally empty condition and totally full condition can reach about 7 mm [3]. Based on that, it is believed that in a short time (daytime < 12 h), the fluctuation of GS measurement should all entirely attributable to temperature uncertainty. 

The ratio of error reduction is calculated by dividing the estimated temperature uncertainty at one moment by the difference between the daytime GS measurement reading at the same moment and the nighttime GS measurement reading. The overall temperature uncertainty reduction ratio calculated by the proposed algorithm is larger than the LSM result, as seen in Table 1; this means that the proposed algorithm can grasp the complex relationship better than the LSM method. 

Table 1 reveals the superior performance of our algorithm in all nine sensors, particularly in sensor #4, where we can observe an increase of approximately 30.4%. This demonstrates the significant advantage of our algorithm. Moreover, when considering the average performance across all sensors, our algorithm consistently outperforms the competition. The data in Table 1 allow for a separate comparison of sensor performance. On average, our algorithm achieves a remarkable temperature uncertainty reduction of approximately 87.4%, surpassing the LSM’s performance of 65.5% by an impressive 21.9%.

The improvement can be attributed to the intricate relationship between temperature and temperature uncertainty. While the basic distribution of the GS is evenly spread around the linear fitting line, it varies between different sensor probes. As shown in Figure 6, a simple linear relationship cannot adequately capture the mapping relationship. However, with the use of ANNs, the coefficients can be fine-tuned and the calculated mapping relationship can be improved with a better fitting line, resulting in a more accurate expression. As depicted in the figure, our method shows a better fit to the relationship compared to the LMS method. This is the primary reason for the higher error reduction ratio observed in Table 1.

As shown above, the GS results are distributed evenly around the straight fitting line of the LMS method. However, the fitting line can only roughly grasp the approximate inclination. There are still large distances between the dots and the fitting line. In contrast, our method of applying the ANN can capture the tendency more vividly and more accurately to the dots. The distance between the dots and green dash dots is smaller. This means that our method can grasp the tendency more accurately than the LMS method. This means that our method can predict the GS more accurately by using temperature change information and the established model built by our method.

In practice, GS monitoring is a spatial comparison between sensors at one moment, as illustrated in Equation (3) above. The data for nine sensors are inputted to the trained neural networks and LSM methods separately. The temperature uncertainty is reduced from the raw data by two methods. The overall curves of the raw data, LSM data, and our algorithm data are presented in Figure 7.

In daytime, the temperature uncertainty in measurement can even reach 50 mm. In the afternoon, under solar irradiation from the west, most sensors on the west side are warmed up (except sensor #1(in the shade area of a neighboring oil tank)), as seen by combining location information from Figure 2c.

In areas exposed to direct sunlight (such as sensors #2, #3, #4, #5, and #8), the measured results show larger values. Conversely, in shaded areas (such as sensors #1, #6, #7, and #9), the measured results are smaller. By considering the information from Table 1 and Figure 6, it is evident that the proposed algorithm performs better under larger temperature uncertainties (greater fluctuation). We can observe that the proposed algorithm produces a smoother overall line. This indicates that our algorithm effectively reduces temperature uncertainty across all sensors, thereby minimizing the impact of temperature-induced variations and allowing for meaningful comparisons between all sensors. 

While the overall trend line still exhibits fluctuations for the sensors, it is important to note that the temperature uncertainty is confined within a range of 10 mm. Although this may be considered relatively high for a large scale oil tank, it is still valuable. By leveraging the high accuracy of nighttime GS monitoring over an extended period, the daytime GS data within the 10 mm range can also serve a purpose: it can be utilized for deformation alerts or safety accident alarms, providing valuable insights and aiding in preventive measures.

The primary focus of the proposed algorithm was to address the issue of temperature uncertainty, which can impact the performance of optical GS sensors. The accuracy of our algorithm heavily relies on the quality of the data used. Therefore, the future research will concentrate on improving data collection and data cleansing techniques. Furthermore, we aim to gather and analyze a larger dataset (to facilitate more efficient data processing) and align with advancements in machine learning methodologies.

## 5. Conclusions

In this study, we focused on investigating and improving the temperature uncertainty of GS sensors in oil tanks, to enhance the accuracy of GS monitoring. Given the challenges of limited data availability and the complex relationship between temperature uncertainty and temperature variation, a transfer-learning-based deep learning method was introduced. This algorithm consisted of a pre-training phase and a training phase, which effectively addressed the data scarcity issue and enhanced the generalization ability of the BP neural network. Through practical tests and data comparisons, it was demonstrated that the proposed algorithm achieved a remarkable improvement of up to 21.9% in temperature reduction performance compared to the LSM. This improvement contributes to robust and accurate daytime GS monitoring for oil tanks, thereby enhancing the intelligence and automation of oil tank health monitoring. Moreover, the transfer learning method provides valuable insights for further advancements in measurement systems based on HLS.

## Figures and Tables

**Figure 1 sensors-24-00215-f001:**
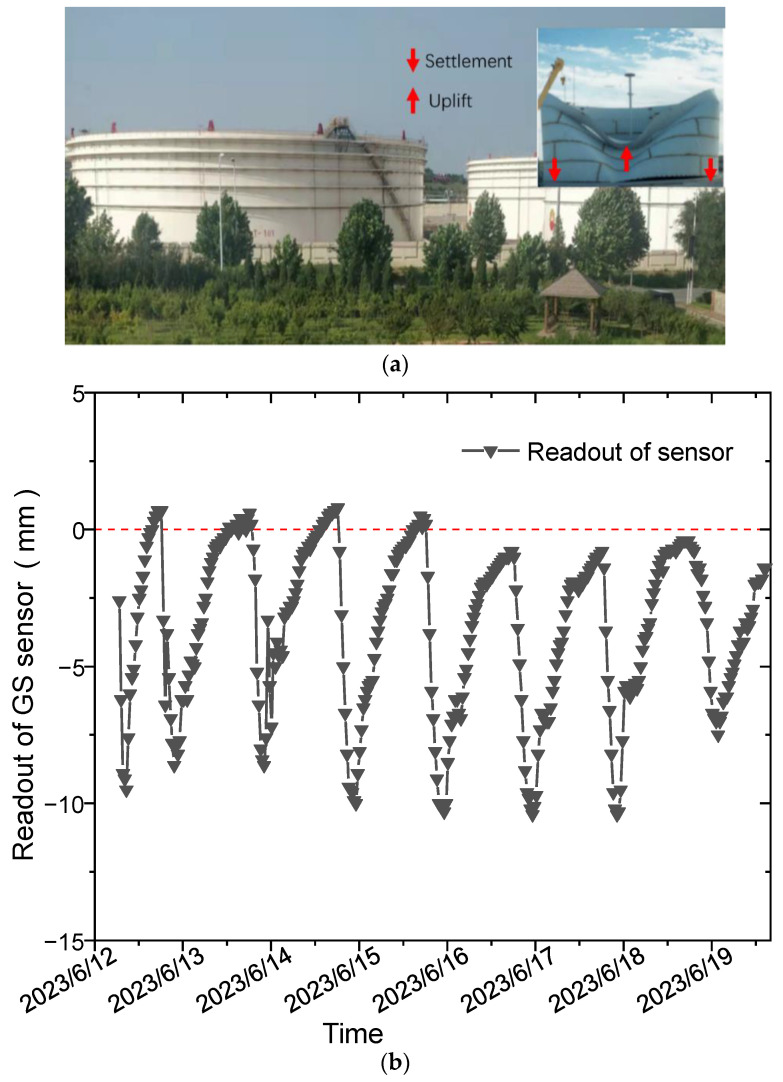
Structural destruction and temperature uncertainty in the GS monitoring sensor of large-scale oil tanks: (**a**) deformation caused by unbalanced GS and (**b**) continuous monitoring results of a GS sensor over a week with a measurement period of one hour.

**Figure 2 sensors-24-00215-f002:**
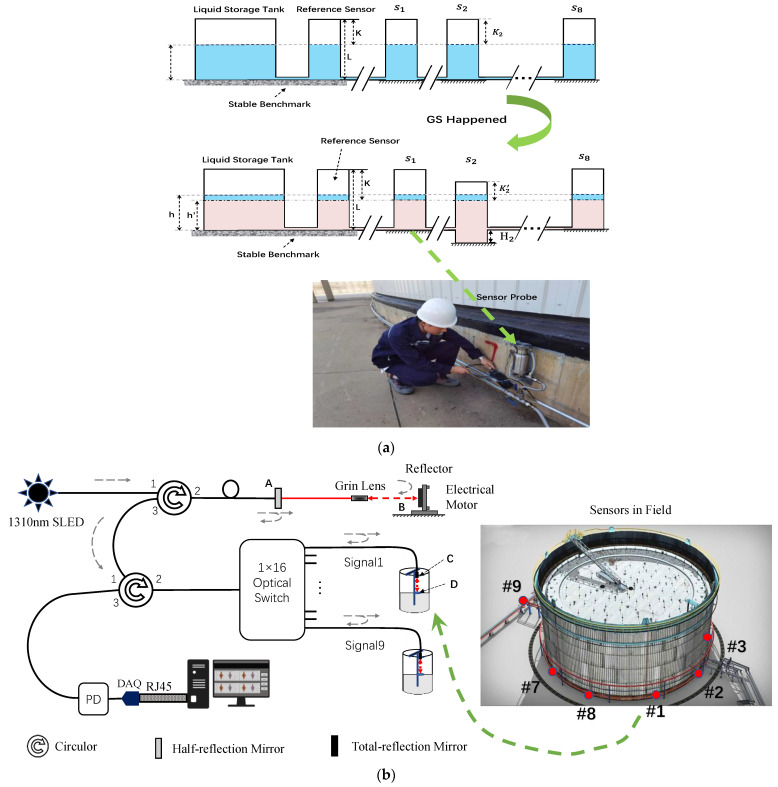
The GS monitoring system scheme based on low-coherence interferometry and HLS: (**a**) the HLS scheme plot and actual sensor installed around the oil tank, (**b**) the optical configuration originating from the sensor installed around the large oil tank and optoelectronic part in control center. In it, A and C denote the half-reflection mirror. B and D denote the total-reflection mirror. 1, 2 and 3 are the ports of circulator. SLED, super-luminescent emitting diode with a central wavelength of 1310 nm; PD, photodetector. DAQ, data acquisition card. RJ45, interface for Ethernet communication. #1 to #9 are index of sensor probes installed around the tank. (**c**) the solar radiation causing unbalanced sunshine. For the different installation positions, different sensors experience different temperatures.

**Figure 3 sensors-24-00215-f003:**
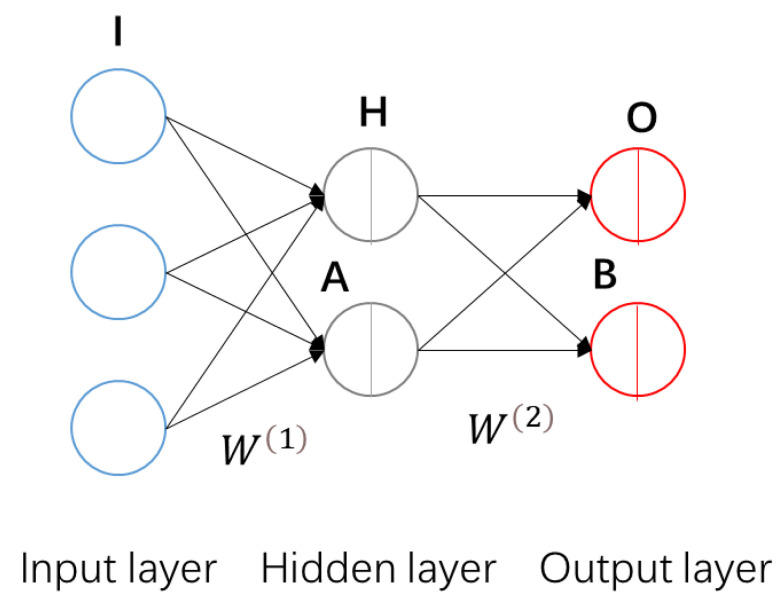
Basic structure of BP neural networks, which consist of three layers: input layer, hidden layer, and output layer. **I**, **H**, **O**, **A** and **B** in the figure are matrices representted the vectors running in the ANN. A and **B** are the bias matrices. **I** is the input matrix. **H** is the matrix of hidden layer. **O** is the output matrix.

**Figure 4 sensors-24-00215-f004:**
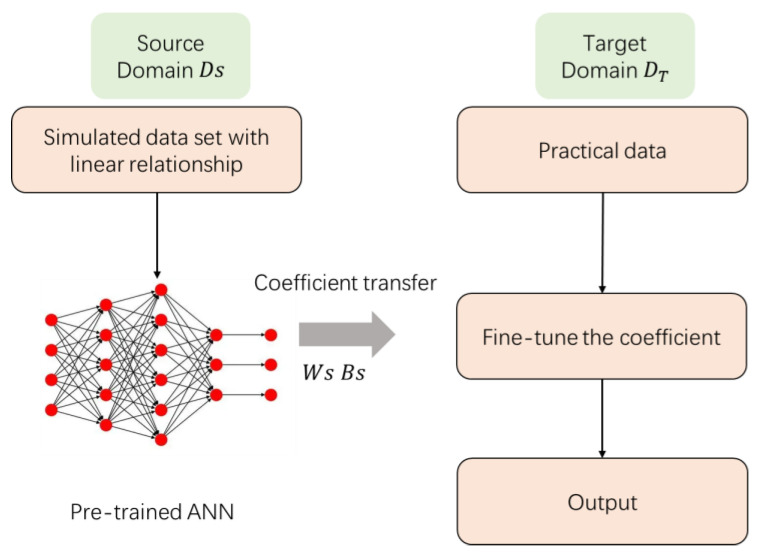
Schematic diagram of transfer-learning-based temperature uncertainty ANN training process. It contains of two training stages, depicted on the left and right sides, which utilize different datasets.

**Figure 5 sensors-24-00215-f005:**
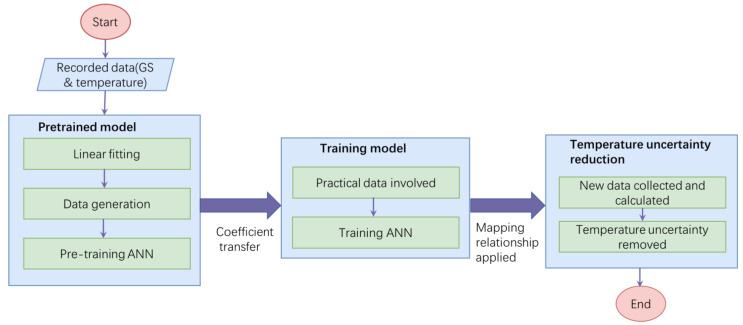
Flowchart of the temperature-reduction algorithm, including pre-training, training and temperature uncertainty reduction components.

**Figure 6 sensors-24-00215-f006:**
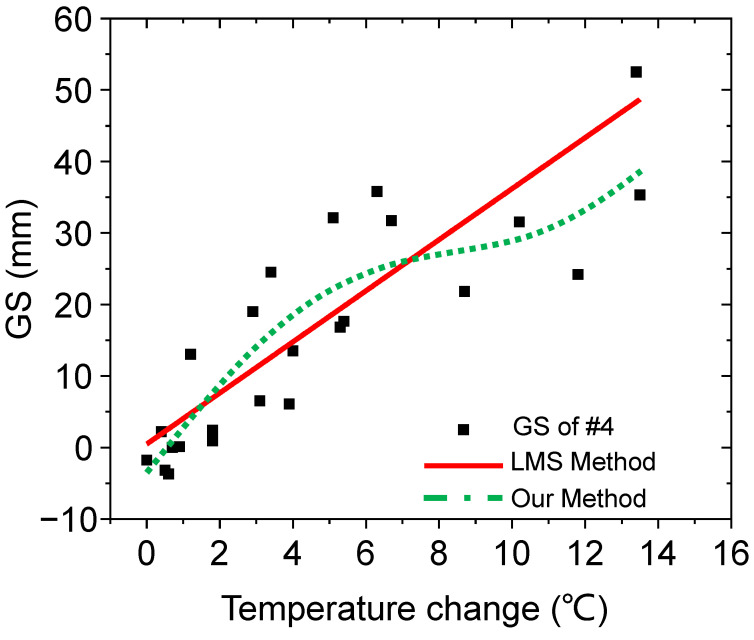
Comparison of GS raw data versus temperature change for sensor #4, along with the fitting lines from the LMS and our method.

**Figure 7 sensors-24-00215-f007:**
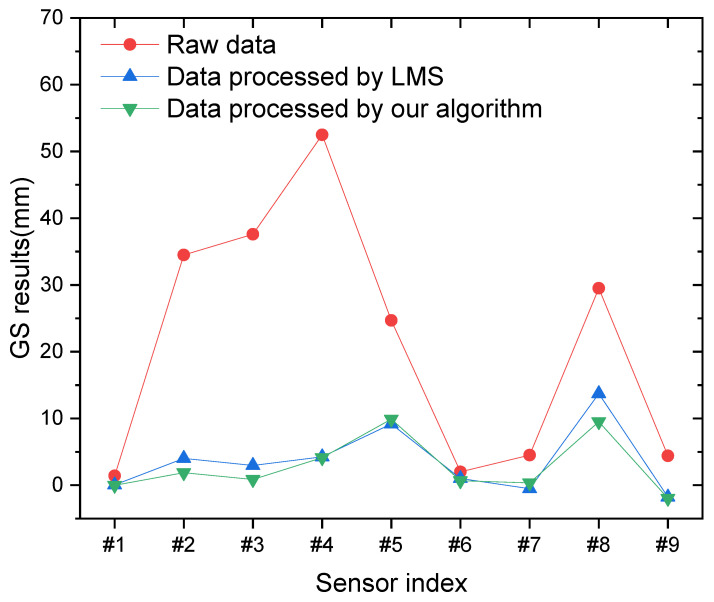
The results from sensors consisting of raw GS measurement results and data processed by the algorithm in different sunshine conditions.

**Table 1 sensors-24-00215-t001:** The ratio of error reduction between the two methods over nine sensors.

Sensor	Ratio of Error Reduction	
	LSM	Proposed Algorithm
#1	70.8%	94.7%
#2	57.3%	87.9%
#3	66.1%	88.2%
#4	59.9%	90.3%
#5	61.5%	81.6%
#6	75.4%	83.5%
#7	68.2%	86.8%
#8	62.7%	89.4%
#9	67.6%	84.1%
Average	65.5%	87.4%

## Data Availability

The data generated or analyzed as part of the research are not publicly available. This research is continuing and the data will be disclosed with the permission of the oil-tank-running corporation.

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
