# Peer review of "Transfer-Learning-Based Temperature Uncertainty Reduction Algorithm for Large Scale Oil Tank Ground Settlement Monitoring"

_sensors, 2023, doi:10.3390/s24010215_

Round 1

Reviewer 1 Report

Comments and Suggestions for Authors

For ground settlement (GS) monitoring sensors used in oil tanks, they are susceptible to nonuniform temperature fields caused by uneven sunshine exposure. This disparity in environmental conditions can lead to errors in sensor readings. To address this issue, this study aimed to analyze the impact of temperature on GS monitoring sensors and establish a mapping relationship between temperature uncertainty (fluctuations of measurement caused by temperature variation) and temperature variation.This outcome highlights the algorithm's potential in enhancing the performance of GS sensors in daytime monitoring and contributing to the safe operation of oil tank facilities and infrastructure health monitoring. However,a detailed description of Figures 2 and 4 is still missing. Lack of explanation, analysis, and discussion of Figure 6.

Comments on the Quality of English Language

no.

Reviewer 2 Report

Comments and Suggestions for Authors

The article focuses on the temperature factors that are widespread in the influence of environmental errors, and using algorithms to study temperature-induced uncertainty is innovative, The starting point of the article is relatively novel, The article has a rich and complete content. However, there are still some shortcomings in the article.

1.       The article mentioned in the introduction of large size tank block the sun light and temperature difference, it is obvious, but can reach 30℃ temperature difference, remain skeptical, temperature is the main research object in this article, Suggestions for the temperature difference gives more specific description, such as the source of the extreme temperature difference and possible region, etc. Because the study is based on the uncertainty of temperature, and when the temperature difference is biased, the uncertainty caused by the studied temperature may not exist directly.

2.       The question of the mapping relationship between temperature and temperature uncertainty studied in the article is too straightforward, and the content of analysis and elaboration is insufficient, which gives people feel that a research question is directly raised, rather than the problem raised by gradually analysis. And there is no analysis why the mapping between temperature and temperature uncertainty can solve the problem of sensor affected by temperature.

3.       The size of data points in Figure 1 (b) is not suitable for the curve thickness. It is suggested to adjust the size of data points to make the data curve image more beautiful.

4.       In the introduction mentions the limitation and lack of data, and how to ensure the reliability of limited data under the premise of the existence of data, because it is suggested that the learning algorithm will be processed based on these limited data, so the source and reliability of these data need to be guaranteed, and it is suggested to add relevant elaboration content.

5.       The photos taken in Figure 2 (a) and (b) are not coordinated with the overall image composed of the image drawn by the arrow. It is suggested that the layout can be adjusted appropriately to make the whole image more coordinated.

6.       To simplify the content of BP neural network in artificial neural network. The article should focus on the description of what you do, rather than spend a large amount of time describing the definite details of existing knowledge.

7.       The description of the engineering background in the transfer learning foundation is too simple. The algorithm studied in this paper is the detection sensor in the service and engineering.

8.       The algorithm flow chart is too simple, and the layout needs to be refined and improved. It is suggested to lay out the algorithm flow chart.

9.       9. The elaboration of the flow chart of the temperature reduction algorithm is the key content of the article. The discussion language should avoid colloquial and increase the academic nature. It is suggested to further refine and condense the content of the description book.

10.   In the description of the actual data and the test results in the conclusion, there is a lack of detailed description of how to obtain effective experimental test data. It is suggested to increase the description content appropriately to enhance the reliability of the comparison conclusion.

11.   In the actual test and data comparison, the proposed algorithm allows the comparison of meaning between all sensors. How this is guaranteed and achieved. It is recommended to add relevant elaboration.

Comments on the Quality of English Language

The Quality of English Language should modify.

Reviewer 3 Report

Comments and Suggestions for Authors

(1)In Artifical neural network. How to obtain reasonable  weight?

(2)The introduction of the engineering background is unclear. Please provide a detailed introduction

(3)All the cited references are relevant to the research, but I don’t think the introduction has provide sufficient background and include all relevant references. Please refer to and cite the following references to improve it:

Effects of land use classification on landscape metrics based on remote sensing and GIS

Shear-related roughness classification and strength model of natural rock joint based on fuzzy comprehensive evaluation.

A new comprehensive evaluating method for assessing the sustainability credentials of the central air-conditioning system

Application Research of a Biomass Insulation Material: Eliminating Building Thermal Bridges

Comments on the Quality of English Language

 the Quality of English Language should be impoved.
